# Reasoning Makes Good Annotators : An Automatic Task-specific Rules Distilling Framework for Low-resource Relation Extraction

**Yilin Lu**
Zhejiang University
22121281@zju.edu.cn

**Juncheng Li**
Zhejiang University
junchengli@zju.edu.cn *

**Xiaoqiang Wang**
Zhejiang University
xq.wang@zju.edu.cn

**Haochen Shi**
Université de Montréal
haochen.shi@umontreal.ca

**Tao Chen**
Zhejiang University
ttc@zju.edu.cn

**Siliang Tang**
Zhejiang University
siliang@zju.edu.cn

## Abstract

Relation extraction is often challenged by insufficient labeled data. Previous methods exploit knowledge from unlabeled data by generating pseudo labels in a self-training pipeline, which suffers a gradual drift problem. Logic rules, a transferable and explainable form of expert knowledge, have achieved promising success by improving the model with weak labels. But manually writing comprehensive rules set is challenging and tedious. To alleviate the human labor of writing high-quality rules, in this work, we propose **ARIA**, an **A**utomatic task-specific **R**ules d**I**stilling fr**A**mework. Specifically, we guide the pre-trained language model to reason rules as experts and compose them into robust compound rules for data labeling. Besides, **ARIA** could continuously enrich the rules set to power the labeling ability by discovering reliable model-labeled data for distinguishable rules generation. Experiments on two public datasets demonstrate the effectiveness of **ARIA** in a low-resource scenario.

## 1 Introduction

Relation extraction is a fundamental task in natural language processing. Training supervised models with manually annotated data is labor-intensive. This motivates methods for model learning under a low-resource setting with limited annotations.

Semi-supervised methods (French et al., 2018; Sun and Grishman, 2012) aim to explore knowledge from the unlabeled data for better model generalization. Self-training pipeline (Rosenberg et al., 2005; Lin et al., 2019) iteratively adds the model's high-confidence predictions over unlabeled set to the training set and re-trains the model. However, the noise in the model-labeled data may accumulate during the training process (gradual drift problem).

The logic rule is an explainable and transferred form to summarize knowledge, which could replace human for weak labels generation. Since the

---

Juncheng Li is the corresponding author.

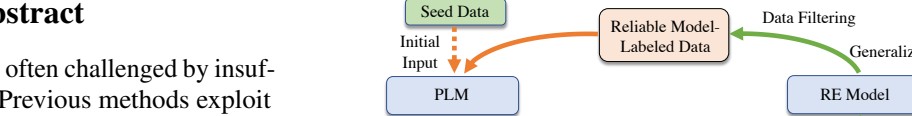

Figure 1: An illustration of how **ARIA** continuously explore PLM-reasoned rules for model improvement.

human written rules (Zhou et al., 2020) are time-consuming and difficult to be completed enough for emerging domains, some work attempts to generate logic rules automatically. For example, the distant supervised methods (Riedel et al., 2010) extract the knowledge base (KB)'s facts as rules for data labeling. These methods label the sentences containing the specific entity pair with KB's relation label regardless of the context, which generates noise labels easily. Thus, how to handle the data's context for accurate labeling deserves studying.

Recently, the pre-trained language model (PLM) shows broad cognitive capabilities that could be distilled into downstream tasks and work well even without any training data (Kojima et al., 2022). Specifically, some work (Wei et al., 2022; Saparov and He, 2022) propose the chain of thought prompts to exploit the PLM's reasoning ability by guiding it to generate the intermediate natural language reasoning process like human. The explainable reasoning process infers the association between the input and output and could be deemed as the prerequisite for the output answer. Motivated by this, given labeled instances as input, we guide the PLM to imitate human relation reasoning manner and summarize the key information supporting relation inference from the reasoning process into transferred rules automatically.

We propose an automatic task-specific logic rules distilling framework, **ARIA**, which leverages PLM to replace human for continuously high-quality labeling rules discovery. As shown in Figure 1, starting from limited seed data, ARIA alter-

nates between Data-to-Rule stage and Rule-to-Data stage. The former guides the PLM to reason specific rules from labeled data by following human reasoning paths and asks the PLM to merge them into compound rules. The latter adopts the rules over the unlabeled set to improve the RE model, which is leveraged to generalize beyond the existing rules. Then we filter reliable model-labeled data for further high-quality rules generation. Different from the previous work, which summarizes rules based on the restricted knowledge from the experts or knowledge bases, **ARIA** could continuously explore comprehensive rules by guiding PLM to imitate human reasoning manner.

There are two major challenges to automatically generate task-specific rules for accurate labeling: 1) how to guide the PLM to follow human's reasoning manner and summarize crucial information of relation inference into comprehensive rules; 2) how to diversify the reasoned rules set with high-quality rules to improve the labeling ability.

To solve the first issue, we guide PLM to imitate typical reasoning chains of human (e.g., induction, abduction) by defining several types of meta-rule templates, which derives the key information related to the relation inference that can be used for the rules construction. For each data, to summarize rules from it, we guide the PLM's reasoning by prompts, which are built by following the reasoning-specific meta-rule templates, and the output reasoning words are used to build different types of rules. Since some work (Wang et al., 2022) shows reasoning in different ways helps answer correctly, we compose the reasoning rules into compound rules and ask PLM to pick out the most comprehensive compositions for robust labeling.

To solve the second issue, the RE model improved by the rule-labeled data, is used to generalize beyond the existing rules and alleviate the gradual drift problem. Since the PLM's reasoning rules conclude the information crucial for relation inference, for each relation, we pick the model-labeled data that could generate rules consistent with this relation's existing rules and distinguishable from other relations' by modeling their rules' relevance. Specifically, we propose a *Graph-based Data Filter*, which builds a graph of both model-labeled data and seed data to propagate their rules' features. For each relation, the model-labeled data with features close to its seed data and far from others' are picked for further rules generation. Compared with

previous work, our method leverages PLM's broad knowledge rather than human knowledge or the restricted knowledge of jointly trained modules to discover data for high-quality rules generation.

In summary, our contributions are four-fold:

1. We develop a framework, **ARIA**, to guide the PLM to continuously summarize comprehensive labeling rules following different reasoning paths as human.

2. We propose a *Graph-based Data Filter* that leverages PLM's knowledge to discover reliable model-labeled data to generate distinguishable rules and enrich the rules set for accurate labeling.

3. **ARIA** achieves a great performance on two public datasets TACRED and SemEval under the low resource setting, which outperforms all the baselines and on average achieves 2.65% higher F1 than the best one.

4. We discuss **ARIA**'s potential under ChatGPT. Embedding Roberta as a safe and efficient annotator, **ARIA** gets competitive labeling precision as ChatGPT. We also show the small-scaled PLM's potential to assist ChatGPT for reliable reasoning: the in-context learning enhanced by our rules' representative information could reduce ChatGPT's hallucination and improve at most 23.04% in precision.

## 2 Related Work

**Semi-Supervised Methods.** With a low requirement of labeled data, semi-supervised methods (French et al., 2018; Sun and Grishman, 2012; Li et al., 2020, 2023; Yu et al., 2023) exploit knowledge from the unlabeled set. Since self-training methods (Rosenberg et al., 2005) accumulate noise in pseudo labels (Hu et al., 2021b), some methods (Han et al., 2018; Lin et al., 2019) lower the noise by a joint training dual retrieval module, whose classification ability is limited by the training data' scale. Our work, instead, denoises by a Data Filter, which leverages PLM's broad cognitive capabilities to pick reliable model-labeled data for distinguishable rules generation.

**Weakly Supervised Methods.** Logic rules are proposed to improve the model with weak labels. Since the manually written rules are expensive and difficult to be completed enough for emerging domains (Zhou et al., 2020), many works discover rules automatically from KB (Riedel et al., 2010;

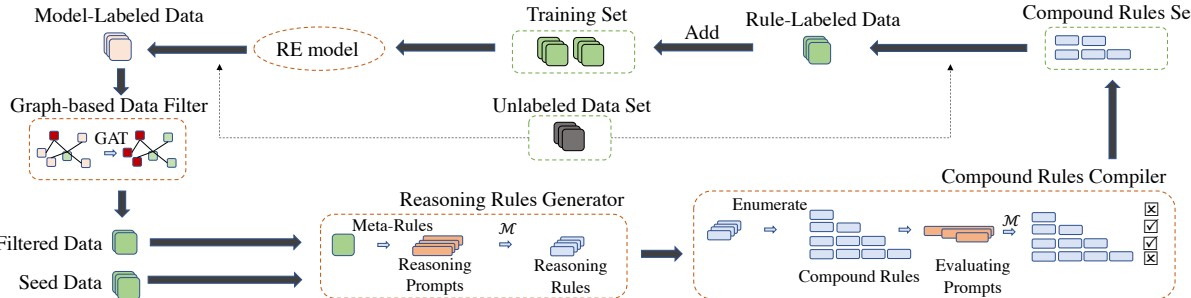

Figure 2: The overall framework of **ARIA**. In each iteration, **ARIA** 1) leverages the *Reasoning Rules Generator* to imitate human's reasoning manner and summarize rules from the labeled data; 2) asks the *Compound Rules Compiler* to compose reasoning rules into robust compound rules; 3) utilizes compound rules to label data for training set enrichment; 4) learns a RE model to predict on the unlabeled data and uses *Graph-based Data Filter* to select reliable model-labeled data for further rules generation in the next iteration.

Ye and Ling, 2019). Recently, language models has shown their ability on various tasks based on prompting (Li et al., 2022a,b, 2021; Lu et al., 2023). Since the scale of KB is limited, PRBOOST (Zhang et al., 2022) asks PLM by prompt to predict the relation directly and take the predictions as rules. This rule construction manner lacks capturing the task-specific reasoning process, making the rules less transferred and explainable. Instead, **ARIA** builds rules in a fine-grained manner by guiding PLM with different reasoning paths. Besides, PRBOOST requires human for rules selection, while **ARIA** could automatically pick the data that could generate high-quality rules by modeling their reasoning rules' dependency.

## 3 Method

We introduce **ARIA** in this section. Given limited seed data, **ARIA** continuously distills specific rules by PLM for labeling over the unlabeled set and improves the task model with rule-labeled data.

**Overview.** As shown in Figure 2, our framework iterates among the four steps: 1) **From Data to Rules:** Given a set of labeled data, we propose a *Reasoning Rules Generator* to guide the PLM to generate specific reasoning rules of different reasoning path from each labeled data. 2) **From Simple Rules to Compound Rules**: A *Compound Rules Compiler* is proposed to compose each data's reasoning rules and ask the language model to pick the comprehensive composition as compound rules. 3) **From Rules to Data**: The compound rules are used to generate labels over the unlabeled set for further RE model improvement. 4) **From Data to Data**: The rule-labeled data are merged with seed data to train a RE model, which is adopted to predict the unlabeled data. The *Graph-based Data*

*Filter* then selects the reliable model-labeled data for further rules generation. Finally, the filtered data are taken as the input for the next iteration.

Notice that in step 4) we pick the model-labeled data rather than rule-labeled data for further rules discovery, which are less likely to accumulate repeating patterns in the rules set based on the model's generalization ability. For initialization, the seed data are taken as the input in the first iteration. Then for the iteration $t + 1$, the input labeled data is the filtered data output from the iteration $t$.

### 3.1 Reasoning Rules Generator

In this section, we introduce the design of our meta-rule templates, which are instantiated by building cloze-style prompts to guide the PLM to reason specific rules under different reasoning paths.

**Meta-Rule Templates Designing.** We propose four meta-rule templates to build reasoning prompts from each instance, as shown in Table 1, where [input] indicates the sentence, [A], [B] denotes the mention of head and tail entities and [M] is the mask token. The design of the meta-rule templates follows four types of reasoning in Causality, which aims to guide the reasoning of the induction, abduction, intervention and counterfactual between entities expressed in the sentence.

To be more specific, 1) **Induction Meta-Rule** treats one entity as a premise and puts mask tokens for reasoning the conclusion related to the other entity. 2) **Abduction Meta-Rule** treats one entity as a conclusion and aims at reasoning the premise related to the other. 3) **Intervention Meta-Rule** aims at reasoning the intervention brought by one entity to the other. 4) **Counterfactual Meta-Rule** guides to reason what will happen to one entity if the other entity does not exist.

| Type | Meta-Rule Template |
|------|-------------------|
| Induction | [Input]. We can infer that, [A] leads to [B]'s [M][M], while [B] leads to [A]'s [M][M]. |
| Abduction | [Input]. We can infer that, [A] is a [M] of [M] for [B], while [B] is a [M] of [M] for [A]. |
| Intervention | [Input]. We can infer that, [A] brought [M] to [B], while [B] brought [M] to [A]. |
| Counterfactual | [Input]. We can infer that, without [A], [B] will [M][M], while without [B], [A] will [M][M]. |

Table 1: Meta-rule templates to build reasoning prompts. [input], [A] and [B] refer to the sentence, head entity and tail entity. [M] is the mask token.

**Reasoning Rules Instantiation.** As shown in Figure 3, given an instance, to generate specific rules from it, we put its sentence and entity mentions to the slots in the meta-rule templates to get four prompts. Then each prompt is fed to PLM respectively and the most likely predicted words for mask tokens are taken as reasoning words. Instantiated rules are obtained by filling all the slots (including mask tokens) in meta-rule templates with specific information. Thus, each data could get four reasoning rules, denoted as $\{p_i\}_{i=1}^4$, with the corresponding reasoning words $\{w_i\}_{i=1}^4$.

In Sec. 4.6, case studies are conducted to show more examples of the reasoning rules.

### 3.2 Compound Rules Compiler

Some work (Wang et al., 2022) has shown reasoning in different ways is beneficial for finding the correct answer. Thus, for each data, we compose its four reasoning rules into robust compound rules.
**Composition Evaluation.** As shown in Figure 3, we ask the PLM by prompt to evaluate if each composition has enough information to figure out the corresponding relation. Given a set of reasoning rules $\{p_i\}_{i\in S}$, the template to build evaluating prompt for this composition is:

*[Input]. $\{p_i'\}_{i\in S}$ Question: Can we infer [A] [Relation] [B]? Answer: [M] (Yes or No).*

where [Input], [A], [B] are the sentence and two entities' mentions in the data, [M] refers to the mask token and [Relation] denotes the data's relation label. $p_i'$ is the mention of reasoning rules $p_i$, with the sentence and the statement "We can infer that" being removed to ensure the prompt's fluency.

After feeding the evaluating prompt to PLM, the probability of the mask token being predicted as 'Yes' is taken as the composition's evaluation score. For each data, we enumerate all the possible compositions of its four reasoning rules and rank them by the evaluation score descendingly. Finally, in each data, the top $N_c$ compositions are picked and each composition will form a compound rule.
**Compound Rule Definition.** Once a compound rule denoted as $p_S$ is built by a composition of

reasoning rules from an instance, it contains the following components: a set of reasoning rules $\{p_i\}_{i\in S}$, a label $l$ (same as the instance's), a threshold $TH$, and a similarity function $g(\cdot, \cdot)$. Given an unlabeled sample $u$, $u$ is matched by $p_S$ if the overall similarity of the reasoning words between $u$ and $p_S$ exceeds the corresponding threshold. Formally,

$$1(u \text{ matched } p_S) = 1\big(g(p_S, u) \geq TH\big) \quad (1)$$

$$g(p_S, u) = \sum_{i\in S} s(w_i, w_i^u) \quad (2)$$

where $w_i$ denotes $p_i$'s reasoning words. Notice we extract $u$'s reasoning words $w_i^u$ by following the same manner as $w_i$ introduced in Sec. 3.1.
**Similarity Measurement.** Since the reasoning words are predicted by the PLM from a large vocabulary, the hard matching manner may lead to a low rule coverage. Thus, we design a soft matching manner, given the reasoning words of a rule and unlabeled data, we compute their embedding and take the cosine similarity as the matching score:

$$s(w_i, w_i^u) = Cos(e(w_i), e(w_i^u)) \quad (3)$$

**Reasoning Words Embedding.** For rules or unlabeled data, we embed each word in $w_i$ by GloVe (Pennington et al., 2014) and concatenate the embedding as the reasoning words' embedding $e(w_i)$.
**Data Labeling & Training Set Enrichment.** Given an unlabeled data $u$ and a compound rules set $R$, to create the rule-generated label, we go through $R$ and measure if $u$ matches each compound rule. When $u$ is matched by multiple compound rules with conflicting labels, the label with the largest similarity $g$ is chosen to label $u$ and this largest similarity is taken as the labeling confidence.

In each iteration, the top $N_d$ rule-labeled data with the highest labeling confidence are selected to enrich the training set.

### 3.3 Graph-based Data Filter

We utilize the RE model learned on the training set to generalize over the unlabeled set and provide model-labeled data for new rules discovery.

For each relation, to select the model-labeled data having inference features consistent with its

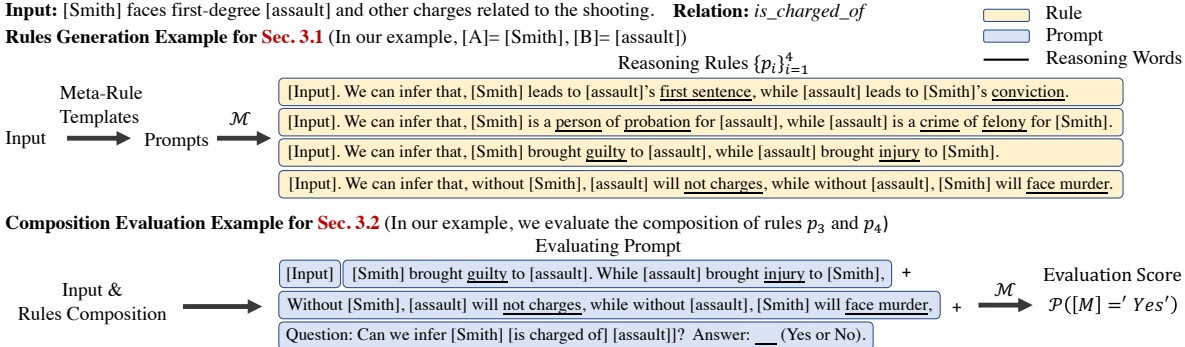

**Input:** [Smith] faces first-degree [assault] and other charges related to the shooting.   **Relation:** *is_charged_of*
**Rules Generation Example for Sec. 3.1** (In our example, [A]= [Smith], [B]= [assault])

□ Rule
□ Prompt
— Reasoning Words

Reasoning Rules $\{p_i\}_{i=1}^4$

Meta-Rule Templates → Prompts $\mathcal{M}$

Input →

[Input]. We can infer that, [Smith] leads to [assault]'s first sentence, while [assault] leads to [Smith]'s conviction.
[Input]. We can infer that, [Smith] is a person of probation for [assault], while [assault] is a crime of felony for [Smith].
[Input]. We can infer that, [Smith] brought guilty to [assault], while [assault] brought injury to [Smith].
[Input]. We can infer that, without [Smith], [assault] will not charges, while without [assault], [Smith] will face murder.

**Composition Evaluation Example for Sec. 3.2** (In our example, we evaluate the composition of rules $p_3$ and $p_4$)

Evaluating Prompt

Input & Rules Composition →

[Input] | [Smith] brought guilty to [assault]. While [assault] brought injury to [Smith], | +
Without [Smith], [assault] will not charges, while without [assault], [Smith] will face murder, | +
Question: Can we infer [Smith] [is charged of] [assault]]? Answer: ___ (Yes or No).

$\mathcal{M}$ → Evaluation Score $\mathcal{P}([M] =' Yes')$

Figure 3: Examples for generating reasoning rules and evaluating rules composition given the input sentence "Smith faces first-degree assault and other charges related to the shooting".

seed rules and far away from other relations', we build a graph for this relation by connecting the model-labeled data with seed data based on their reasoning words' relevance. Then a classifier is trained to figure out if a data belongs to this relation (Positive) or not (Negative).

**Initial Node Embedding.** Since a node represents a data, for each node, we concatenate the data's four types of reasoning words' embedding (introduced in Sec. 3.2) as its initial node embedding.

**Graph Construction.** We build a $k$-nearest neighbor graph $G_r = \{V = V_p \bigcup V_s, A\}$ for relation $r$, which connects nodes based on their initial embedding's cosine similarity and aims to model the data's inference features relevance. $V_p$, $V_s$ are nodes for model-labeled data and seed data, $A$ is the adjacency matrix. To get the binary supervision signals, each node in $V_s$ will get a Positive binary label if its data belongs to $r$, otherwise, Negative.

**Inference Features Propagation.** For relation $r$ and its corresponding graph $G_r$, we apply an independent 2-layer graph attention network (Veličković et al., 2018) to propagate nodes' inference features. Given node $i$'s representation $h_i$ and its neighbors $N_i$, in each layer the attention weight between node $i$ and $j$ can be computed as:

$$a_{ij} = \frac{exp(f(A^\mathsf{T}[Wh_i, Wh_j]))}{\sum_{k \in N_i} exp(f(A^\mathsf{T}[Wh_i, Wh_k]))} \quad (4)$$

where $W$ is parameter matrix and $f$ is LeakyReLu function. Then the $i^{th}$ node representation of the next layer is updated as:

$$h_i^* = a_{ii}Wh_i + \sum_{j \in N_i} a_{ij}Wh_j \quad (5)$$

**Objective.** The classifier's objective is defined as:

$$L = L_{sup} + L_{nei} \quad (6)$$

where $L_{sup} = -\sum (y_i log\mathcal{P}_i + (1 - y_i)log(1 - \mathcal{P}_i))$ $(i \in V_s)$ is the supervised loss, $y_i$ is the bi-

nary label of the node, $\mathcal{P}_i$ is the predicted possibility of node $i$ being Positive, and $L_{nei} = \sum_{j \in V} \sum_{k \in N_j} \|h_j - h_k\|_2$ is to encourage nodes with similar inference features to be close to each other in the representation space.

**High-quality Data Filtering.** To pick the model-labeled data containing inference features consistent with the existing rules for relation $r$ and distinguishable from other relations, we 1) compute the centroids' representation of Positive and Negative seed nodes as the average of corresponding nodes' representation; 2) measure each model-labeled node's cosine similarity with the two centroids' representation as $S_{pos}$ and $S_{neg}$; 3) rank all model-labeled nodes by $S_d = S_{pos} - S_{neg}$ descendingly to find distinguishable nodes. Then we select the top $N_p$ model-labeled nodes (data) for further rules generation of relation $r$.

## 4 Experiments

### 4.1 Datasets

By following (Hu et al., 2021a), we conduct experiments on two public datasets widely used for relation extraction, SemEval (Hendrickx et al., 2010) and TACRED (Zhang et al., 2017). The datasets statistic is shown in Appendix A.1. TACRED is more complicated than SemEval, since it contains more relation types and a larger distribution bias between positive and negative instances.

### 4.2 Baselines

**Supervised Baselines.** We build classifiers with three common encoders and train them with only labeled data: **LSTM** (Hochreiter and Schmidhuber, 1997), **PCNN** (Zeng et al., 2015) and **BERT** (Kenton and Toutanova, 2019). Since BERT performs best, we take it as the encoder of all the following baselines and **ARIA**'s RE model for fair evaluations.

| Methods / %Labeled Data | SemEval | | | TACRED | | |
|---|---|---|---|---|---|---|
| | 5% | 10% | 30% | 3% | 10% | 15% |
| LSTM (Hochreiter and Schmidhuber, 1997) | 22.65 | 32.87 | 63.87 | 28.68 | 46.79 | 49.42 |
| PCNN (Zeng et al., 2015) | 41.82 | 51.34 | 63.72 | 40.02 | 50.35 | 52.50 |
| BERT (Kenton and Toutanova, 2019) | 70.71 | 71.93 | 78.55 | 40.11 | 53.17 | 55.55 |
| Self-Training$_{BERT}$ (Rosenberg et al., 2005) | 71.34 | 74.25 | 81.71 | 42.11 | 54.17 | 56.52 |
| Mean-Teacher$_{BERT}$ (Tarvainen and Valpola, 2017) | 70.05 | 73.37 | 80.61 | 44.34 | 53.08 | 53.79 |
| RE-Ensemble$_{BERT}$ (Lin et al., 2019) | 72.35 | 75.71 | 81.34 | 42.78 | 54.83 | 55.68 |
| DualRE-Pairwise$_{BERT}$ (Lin et al., 2019) | 74.35 | 77.13 | 82.88 | 43.06 | 56.03 | 57.99 |
| DualRE-Pointwise$_{BERT}$ (Lin et al., 2019) | 74.02 | 77.11 | 82.91 | 43.73 | 56.28 | 57.72 |
| MRefG$_{BERT}$ (Li and Qian, 2020) | 75.48 | 77.96 | 83.24 | 43.81 | 55.42 | 58.21 |
| MetaSRE$_{BERT}$ (Hu et al., 2021a) | 78.33 | 80.09 | 84.81 | 46.16 | 56.95 | 58.94 |
| GradLRE$_{BERT}$ (Hu et al., 2021b) | 79.65 | 81.69 | 85.52 | 47.37 | 58.20 | 59.93 |
| **ARIA**$_{BERT}$ **(Ours)** | **80.24** | **82.40** | **86.07** | **49.59** | **60.86** | **62.57** |
| BERT w. gold labels | 86.66 | 87.25 | 87.87 | 62.56 | 64.15 | 64.51 |

Table 2: Micro F1 (%) of methods on SemEval and TACRED datasets with different scales of seed data and 50% unlabeled data. For each method we conduct five runs with random seeds and report the average performance.

**Low-Resource Learning Baselines.** We compare **ARIA** with baselines studied under the low-resource setting. 1) **Self-Training** (Rosenberg et al., 2005) generalizes the learned model on unlabeled set and updates the model with pseudo labels. 2) **Mean-Teacher** (Tarvainen and Valpola, 2017) forms the teacher model from students by encouraging consistent predictions for similar inputs. 3) **DualRE** (Lin et al., 2019) jointly trained a retrieval module to provide pseudo-labeled data for the prediction module. Pointwise or Pairwise is the way to measure the retrieved data's quality. 4) **RE-Ensemble** (Lin et al., 2019) replaces DualRE's retrieval module with a prediction module. 5) **MRefG** (Li and Qian, 2020) connects the unlabeled data with labeled ones semantically to build reference graphs. 6) **MetaSRE** (Hu et al., 2021a) eval pseudo-labeled data's quality by meta-learning from the model's successful and failed attempts. 7) **GradLRE** (Hu et al., 2021b) encourages the pseudo-labeled data to imitate the labeled data's gradient descent direction.

**RE-Gold.** We train our RE model with both seed data and the unlabeled data assigned with the gold labels to indicate the upper bound performance (denoted as BERT w. gold labels).

### 4.3 Implementation Details

**Data Preparation.** To imitate the low-resource scenario, we follow (Hu et al., 2021a) and utilize stratified sampling to divide the original training set into seed data and unlabeled data. Specifically, for SemEval, we sample 5%, 10%, 30% training data as seed data, while for TACRED, 3%, 10%, 15% are sampled. For both datasets, 50% training

data are sampled as unlabeled data, whose gold labels are unavailable during the learning process. **Hyperparameter Setting.** Roberta-base (Liu et al., 2019) is taken as our PLM for rules reasoning. For the *Compound Rules Compiler*, in each data the top $N_c = 8$ high-quality combinations are picked to build compound rules and the threshold $TH$ for data labeling is set to 0.8. In each iteration, the top $N_d = 100$ rule-labeled data with the highest labeling confidence are added to the training data. We take DualRE-Pointwise as our RE model, trained with our training set (containing both seed data and rule-labeled data) and the unlabeled set. For each relation, the *Data Filter* picks the top $N_p = 15$ high-quality model-labeled data for further rules generation in the next iteration.

### 4.4 Overall Performance

Table 2 shows the performance on two datasets. **Overview:** We could observe that **ARIA** outperforms all the baselines on both datasets with different scales of seed data, which demonstrates **ARIA**'s effectiveness and stableness. Specifically, compared with the previous SOTA work GradLRE, **ARIA** average achieves an F1 improvement of 0.62% in SemEval and 2.51% in TACRED.

Compared with the DualRE-Pointwise that only trained with the seed data and unlabeled data, the RE model (with the same structure) embedded in **ARIA** average gets F1 score 4.89% higher in SemEval and 5.10% higher in TACRED. This result shows a significant improvement from the comprehensive rules distilled from the PLM.

Among the two datasets, **ARIA** gets greater im-

**Org: Founded_by**

[Lee, the third son of the late Samsung group founder Lee Byung, became vice chairman of the Samsung Corporation.]
We can infer that, [Samsung] leads to [Lee Byung]'s business success, while [Lee Byung] leads to [Samsung]'s business compony.

**Org: Member_of**

[Shiite List groups lots of Shiite and other parties, including the Badr Organization former Badr Corps and independents.]
We can infer that, [Badr Organization] leads to [Shiite]'s large list, while [Shiite] leads to [Badr Organization]'s Shiite party.

**Cause-Effect (e1, e2)**

[Asthma causes swelling and narrowing of the airways.]
We can infer that, [Asthma] leads to [swelling]'s bad symptoms, while [swelling] leads to [Asthma]'s worsening death.

**Whole-Component (e1, e2)**

[The kitchen holds a cooker, fridge, microwave oven, in short: everything you need to prepare a light meal.]
We can infer that, [kitchen] is a kind of room for [cooker], while [cooker] is a kind of storage for [kitchen].

**Cause-Effect (e1, e2)**

[The burned fuels create air pollution that contributes to global warming and causes respiratory disease.]
We can infer that, [pollution] is a source of risk for [disease], while [disease] is a source of blame for [pollution].

**Org: Subsidiaries**

[Carnival Cruise Lines president Robert Dickinson, will retire at the end of the year, parent Carnival Corp said.]
We can infer that, [Carnival Corp] is a parent of operations for [Carnival Cruise Lines], while [Carnival Cruise Lines] is a subsidiary of operation for [Carnival Corp].

**Per: Cause_of_Death**

[Beverly Sills, obituary Opera legend Beverly Sills succumbed to lung cancer tonight at age 78.]
We can infer that, [Beverly Sills] brought succumbe to [lung cancer], while [lung cancer] brought death to [Beverly Sills].

**Entity-Origin (e1, e2)**

[When the princess left her native land and traveled west to her bridegroom, she carried silkworm cocoons in her headdress. ]
We can infer that, [princess] brought her to [land], while [land] brought happiness to [princess].

**Per: Parents**

[Catherine was born in Brooklyn in 1920, one of three children of Albert and Gertrude Dittmars Roraback.]
We can infer that, [Catherine] brought married to [Albert], while [Albert] brought brith to [Catherine].

**Per: Date_of_Birth**

[Born on Aug. 15, 1925, in a poor neighborhood of Montreal, Peterson got his passion for music from his father.]
We can infer that, without [Peterson], [Aug. 15, 1925] will be meaningless, while without [Aug. 15, 1925], [Peterson] will be void.

**Product-Producer (e1, e2)**

[The steering committee that drafted the document also passed it through two Palestinian GFM organizers.]
We can infer that, without [committee], [document] will not exist, while without [document], [committee] will be exist.

**Org: Subsidiaries**

["Many People see the kibbutz as a corpse," says Shlomo, head of the Kibbutz Studies Centre at Haifa University.]
We can infer that, without [Haifa University], [Kibbutz Studies Centre] will not exist, while without [Kibbutz Studies Centre], [Haifa University] will be exist.

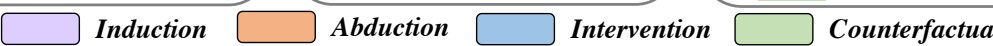

*Induction*   *Abduction*   *Intervention*   *Counterfactual*

Figure 4: Examples of reasoning rules belonging to different relations summarized by the PLM in Section 3.1. The four types of color indicate four types of reasoning paths. The reasoning words for each rule are highlighted.

provements on TACRED compared with all baselines, which shows **ARIA** could better leverage the PLM to exploit high-quality rules to improve the model's learning of more complex relations with skewed data distribution.

## 4.5 Ablation Study

We present ablation studies on SemEval to show the components' effectiveness (Table 3).

| Methods / %Labeled Data | 5% | 10% | 30% |
|---|---|---|---|
| **ARIA** | 80.24 | 82.40 | 86.07 |
| **ARIA** w/o GDF | 76.77 | 80.48 | 84.21 |
| **ARIA** w/o CRC | 77.99 | 78.09 | 84.31 |
| **ARIA** w/o Indu. | 75.65 | 80.87 | 84.65 |
| **ARIA** w/o Abdu. | 74.53 | 79.85 | 84.99 |
| **ARIA** w/o Inte. | 75.82 | 80.16 | 83.28 |
| **ARIA** w/o Cont. | 76.72 | 80.61 | 85.47 |

Table 3: Ablation results of **ARIA** and its variants on SemEval with different scales of seed data.

**Removing Graph-based Data Filter (w/o GDF).**

When extracting rules directly from all the model-labeled data without Data Filter, the F1 scores lower $3.47\% \sim 1.86\%$. The decrease gets larger when running with fewer seed data. Thus when the seed data is quite limited, the Data Filter is necessary to pick reliable data from the unfitted model's predictions for high-quality rules generation.

**Removing Compound Rules Compiler (w/o CRC).** When labeling with single reasoning rules rather than compound ones, the performance gets lower, especially when the seeds' scale turns from 5% to 10%. This is because some relations can not be discriminated by single reasoning paths and more seed rules lead to noise labeling. Thus, Compound Rules Compiler is needed for fine-grained data labeling. The effect of labeling noise gets lighter as the seed data is abundant enough (30%).

**Removing one type of meta-rule each time (w/o Indu, w/o Abdu, w/o Inte and w/o Cont).** The result shows that each meta-rule contributes to **ARIA**.

The lack of each meta-rule causes a sharp performance degradation on small-scaled seed data (5%). This is because the labeling noise caused by the incomplete reasoning paths (as shown in Sec. 4.6) can easily lead to the model's misunderstanding of relations when the correct seed data is limited.

### 4.6 Case Study

Figure 4 shows examples of four types of reasoning rules. We can observe that the rules' reasoning words capture the information crucial for relation inference. For example, given an instance with entity pair $(X, Y)$ and meeting "$X$ leads to $Y$'s business success, while $Y$ leads to $X$'s business company", we can infer that $X$ may be a company greatly contributed by $Y$ and $Y$ is likely to be $X$'s founder. Thus our reasoning words are beneficial to sentence understanding and rules construction.

**Necessity of Compound Rules.** Besides, the reasoning rules provided by single reasoning path is not enough to discriminate different relations. For example, under the Counterfactuals reasoning path, the rules of relation *Product-Producer* and *Org: Subsidiaries* have similar reasoning words. Thus, to label the relation *Org: Subsidiaries* accurately, the Abduction rule is necessary, which indicates the effectiveness of compound rules construction.

### 4.7 ARIA's Potential under ChatGPT

To discuss ChatGPT's influence on **ARIA**, taking 5% training data as seed data and 50% as unlabeled data in SemEval, we design different labeling manners based on ChatGPT and compare their labeling precision on the unlabeled data (as Figure 5 shows). Notice that we sort the annotated results by labeling confidence and compute the precision for the top K results (more details are in Appendix A.2).

**ARIA: Competitive Performance with a Light and Privacy-Protected Annotator.** We replace Roberta with ChatGPT for rules generation and composition in our framework, and label data with compound rules (denoted as ARIA-ChatGPT). When Top K is 5∼45, **ARIA**'s labeling precision is 1.67%∼7.72% higher than ARIA-ChatGPT. This is because the hallucination of ChatGPT may make rules unreliable and lead to fake high-confident labeling results. For example, when the ground-truth relation is "NA" (there is no relation between the two entities), ChatGPT may generate unreasonable rules that seem brilliant, while Roberta, the naive PLM, may answer meaningless words. Compared with ChatGPT, Roberta embedded in **ARIA** could

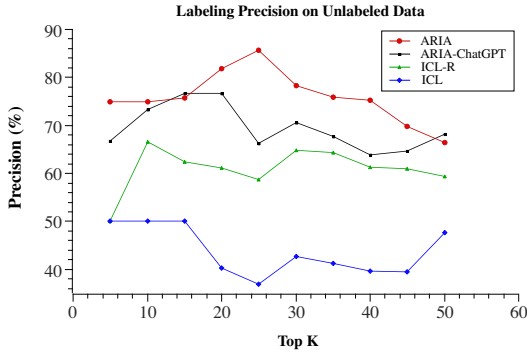

Figure 5: Labeling precision of **ARIA** and different pipelines based on ChatGPT.

predict the reasoning words quickly and is more time-saving to handle large-scale data in real scenarios. Furthermore, Roberta could be deployed privately and avoid the data breach risk. This exploration also shows our framework could embed different types of language models as the annotator.
**Achieve Concentrate In-Context Learning with Rules Knowledge.** We apply in-context learning to annotate the unlabeled data directly by ChatGPT. Taking this as a baseline (denoted as ICL), we design ICL-R, each example data in the context is followed by its reasoning rules mentions instantiated in **ARIA**. We can see ICL-R is at most 23.04% higher than ICL. Since in ICL-R, ChatGPT focuses more on the representative knowledge points for relation reasoning summarized by our rules, it is less likely to be distracted by other information or hallucination caused by the example data. This shows the small-scaled PLM's potential to highlight the key points and assist the reasoning of ChatGPT.
**Future Works: Cooperation for Complex Scenarios.** ChatGPT shows powerful comprehension ability but sometimes generate hallucination. **ARIA** provides an efficient rules-distilling manner that could reduce hallucination, but simple rules matching is not enough for complex semantics. To leverage the advantages of both sides, we could first ask Roberta simple questions and summarize the key points into rules, which are collected to ChatGPT to get a reliable reasoning with less hallucination. We could further ask ChatGPT if its conclusion obeys the rules or if the rules are reasonable, and fix the answer or the rules automatically.

### 5 Conclusion

We propose **ARIA**, which guides the PLM to summarize comprehensive rules as human. Specifically, we build a *Reasoning Rules Generator* to replace human for high-quality rules generation and a *Com-*

*pound Rules Compiler* to compose rules into robust compound rules. Besides, **ARIA** could discover reliable model-labeled data for further rules generation based on the rules relevance. Experiments on SemEval and TACRED show **ARIA**'s effectiveness.

# 6 Acknowledgments

This work has been supported in part by the Zhejiang NSF (LR21F020004), Key Research and Development Projects in Zhejiang Province (No. 2023C01030, 2023C01032), NSFC (No. 62272411), National Key Research and Development Program of China (2018AAA0101900), Ant Group and Alibaba-Zhejiang University Joint Research Institute of Frontier Technologies.

# Limitations

In our *Reasoning Rules Generator*, we take Roberta's most likely predicted words for mask tokens as reasoning words. This may ignore much important information contained in Roberta's predictions. An approach to making better use of Roberta's knowledge is to take Roberta's hidden vectors for the masked tokens as virtual reasoning words. Besides, to better leverage our rules to handle the complex semantics, we could collect the rules with representative key points from Roberta and input them with context to ChatGPT for a reliable answer.

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

# A Appendix

## A.1 Datasets Statistics

In this section we show the statistics of the two public datasets used in our experiments.

| Datasets | Relations | Train | Dev | Test | %Neg. |
|---|---|---|---|---|---|
| SemEval | 19 | 7199 | 800 | 1864 | 17.40 |
| TACRED | 42 | 75049 | 25763 | 18659 | 78.68 |

Table 4: Statistics of the datasets in our experiments. Notice that Neg. refers to the percent of negative instances, which are labeled as *NA*.

## A.2 Implementation Details of ICL, ICL-R and ARIA-ChatGPT

For ARIA-ChatGPT, we redesign cloze templates to prefix templates to exploit ChatGPT's potential. Notice that for both ARIA and ARIA-ChatGPT in Figure 5, we report the labeling precision in the first iteration. For ICL and ICL-R, we randomly select one seed data per relation as example data in context. For each unlabeled data, the ChatGPT predicts 5 times under the contexts with random example data. Then majority voting strategy is

| Inference Type | Meta-Rule Template |
|---|---|
| Induction | [Input]. We can infer that, [A] leads to [B]'s MASK. Question: MASK = ? (Please answer within two words). |
| | [Input]. We can infer that, [B] leads to [A]'s MASK. Question: MASK = ? (Please answer within two words). |
| Abduction | [Input]. We can infer that, [A] is a MASK_1 of MASK_2 for [B]. Question: MASK_1 = ? (Please answer within one word). |
| | [Input]. We can infer that, [A] is a MASK_1 of MASK_2 for [B]. Question: MASK_2 = ? (Please answer within one word). |
| | [Input]. We can infer that, [B] is a MASK_1 of MASK_2 for [A]. Question: MASK_1 = ? (Please answer within one word). |
| | [Input]. We can infer that, [B] is a MASK_1 of MASK_2 for [A]. Question: MASK_2 = ? (Please answer within one word). |
| Intervention | [Input]. We can infer that, [A] brought MASK to [B]. Question: MASK = ? (Please answer within one word). |
| | [Input]. We can infer that, [B] brought MASK to [A]. Question: MASK = ? (Please answer within one word). |
| Conterfactual | [Input]. We can infer that, without [A], [B] will MASK. Question: MASK = ? (Please answer within two words). |
| | [Input]. We can infer that, without [B], [A] will MASK. Question: MASK = ? (Please answer within two words). |

Table 5: Prefix Meta-Rule Template applied in ARIA-ChatGPT. [input], [A] and [B] refer to the sentence, head entity and tail entity.

taken to pick out the final prediction and the max votes number is taken as the labeling confidence.

**Prefix Template for ARIA-ChatGPT.** To generate and compose the rules by ChatGPT, we transfer the cloze template into prefix template by asking ChatGPT what the "MASK" refers to in natural language. The prefix Meta-Rule Template is shown in Table 5. The prefix template for building the evaluating prompt is shown as follows:

*[Input]. $\{p_i'\}_{i \in S}$ Question: Can we infer [A] [Relation] [B]? Answer: (Yes or No).*

> $[Input]_1$
> Q: What is the relation label of $[A]_1$ and $[B]_1$ ?
> A: $[Relation]_1$
>
> …
>
> $[Input]_n$
> Q: What is the relation label of $[A]_n$ and $[B]_n$ ?
> A: $[Relation]_n$
>
> $[Input]$
> Q: What is the relation label of $[A]$ and $[B]$ ? You should answer one of the following relations: $[Relation\ Set]$.
> A:
>
> ICL
>
> $[Input]_1$
> $[P']_1$
> Q: What is the relation label of $[A]_1$ and $[B]_1$?
> A: $[Relation]_1$
>
> …
>
> $[Input]_n$
> $[P']_n$
> Q: What is the relation label of $[A]_n$ and $[B]_n$ ?
> A: $[Relation]_n$
>
> $[Input]$
> $[P']$
> Q: What is the relation label of $[A]$ and $[B]$ ? You should answer one of the following relations: $[Relation\ Set]$.
> A:
>
> ICL-R

Figure 6: Template of contexts in ICL and ICL-R.

where [Input], [A], [B] refers to the sentence and the two entities mentions in the instance and [Relation] donates the instance's relation label mention. $p_i'$ is the mention of reasoning rules $p_i$, with the sentence and the statement "We can infer that" being removed. $S$ represents the set of reasoning rules to be evaluated for composition.

**Context Building in ICL, ICL-R.** The template to build the context is shown in Figure 6. $[Input]_i$, $[A]_i$, $[B]_i$ and $[Relation]_i$ are the sentence, head entity, tail entity and the label of $i^{th}$ example data. $n$ is the number of labels and $[Relation\ Set]$ is the set of all label mentions. $[P']_i = \{p_k'\}_{k=1}^4$ is the set of reasoning rules' mentions of $i^{th}$ example.