# OpenReview forum: "Reasoning Makes Good Annotators : An Automatic Task-specific Rules Distilling Framework for Low-resource Relation Extraction"
_EMNLP/2023/Conference — EMNLP 2023 Findings_

### Official Review · Reviewer_Hf2b · 2023-07-31

**Typos Grammar Style And Presentation Improvements:** 1. The word "donated" used throughout…
**Soundness:** 4

**Excitement:**

3: Ambivalent: It has merits (e.g., it reports state-of-the-art results, the idea is nice), but there are key weaknesses (e.g., it describes incremental work), and it can significantly benefit from another round of revision. However, I won't object to accepting it if my co-reviewers champion it.

**Missing References:**

None

**Paper Topic And Main Contributions:**

In this paper, the authors introduce ARIA, an automatic task-specific logic rules-distilling framework that uses a large language model (LM) to continuously build high-quality rules for labeling relations. In the context of the relation extraction task, the authors argue that neither supervised methods (which require labor-intensive annotations) nor distant supervision approaches (which ignore the data context while assigning relation labels) are suitable for building relation extraction systems under a low-resource setting. Instead, a logic rule, which is an explainable way to summarize knowledge, should be used to generate labels. Humans typically create logic rules; however, ARIA suggests using LMs to build high-quality rules for discovering relations.

ARIA works in the following way: it learns to find high-quality rules for relations from the seed data and annotates a collection of unlabeled data, which is then fed as input to a Relation extraction classifier. The model-predicted relation extraction instances are then used to improve ARIA's generalization capability beyond the rules that it currently knows. For the first step, given a seed-labeled example, ARIA uses a collection of meta-rule templates and an LM to construct a chain of thought reasoning rules. In the next step, one or more previously generated rules are composed to yield a compound rule, which is applied to the unlabeled dataset to identify matching instances. An unlabeled example, u, is said to match a rule r if the similarity score between u and r exceeds some predefined threshold. The top N_d matchings selected from this step are fed to the Relation extraction model to build an updated RE model.

The updated RE model is then applied over the remaining unlabeled set and provides model-labeled data for discovering new rules in the final Graph-based data filtering step. To select model-labeled data consistent with the seed rules, a GAT is trained over the graph built using the seed data and the model-labeled data as nodes and cosine similarity between the nodes as edges. Finally, the filtered data from this iteration is combined with the data from the previous iteration, and the process continues.

The authors use SemEval and TACRED datasets to demonstrate the effectiveness of their approach. Moreover, they illustrate comparisons with existing state-of-the-art approaches and ablation results for the various components of ARIA.

**Questions For The Authors:**

1. In Sec 3.2, L262-263, the paper mentions "... We ask the PLM by prompt to evaluate if each composition has enough information ...." followed by the definition of the Composition prompt; Does this mean that the composition prompt is evaluated for all possible compositions of rules in Table 1? What kind of additional run-time complexity does ARIA incur on this step?

2. The need for Sec 3.2, i.e., the need to instantiate explicit compound rules is unclear. From Equations (1), (2), and (3), we observe that, given an unlabeled sample u, whether u matched p_S (the compound rule) does not depend upon p_S but depends on p_i, i.e., the rules obtained from the meta-rule template and the filled in prompt (from the LM). So, ideally, we donot require an instantiation of the compound rule p_S, correct?

3. A low-resource setting for Relation Extraction task typically occurs in domain specific datasets such as Chemprot [1], SCIERC, etc. and not on Open domain datasets such as TACRED or SemEval. How does ARIA perform on either one of them in comparison to existing SOTA approaches? Also, referring to the evaluation section, using 3% of TACRED (or 5% of SemEval) over five random runs can still have a large variance - depending upon which 3% of TACRED got sampled for training.

4. What is the difference between BERT (Row 3) and BERT w/ gold labels (final row) in Table 2? Any insights into why the performance improvement on TACRED is higher than that of SemEval? Also [2] shows that a significant chunk of the most challenging sentences from the dev/test sets of TACRED are incorrectly labelled, and therefore introduce a newer annotated variant called Re-TACRED with all the fixes. How does ARIA perform on Re-TACRED?

[1] Peng, Yifan, Shankai Yan, and Zhiyong Lu. "Transfer Learning in Biomedical Natural Language Processing: An Evaluation of BERT and ELMo on Ten Benchmarking Datasets." BioNLP 2019 (2019): 58.
[2] Stoica, George, Emmanouil Antonios Platanios, and Barnabás Póczos. "Re-tacred: Addressing shortcomings of the tacred dataset." Proceedings of the AAAI Conference on Artificial Intelligence. Vol. 35. No. 15. 2021.

**Reasons To Accept:**

1. The problem statement is well-motivated and the overall description of the proposed ARIA framework is really ingenious.
2. Improved state-of-the-art results on SemEval and TACRED datasets, followed by a meaningful qualitative and quantitative analysis.

**Reasons To Reject:**

1. The paper needs to be clearer to read and follow. It misses several pieces of information that could make it much more straightforward. For example, L333-336 mentions that the initial node embedding for a data point is built by concatenating the embedding of the data's four types of reasoning words. However, Figure 3 shows that there could be multiple reasoning words for each of the four reasoning templates (Table 1). How does the aggregation take place, then?

2. Specific modules of the proposed ARIA framework are introduced without motivating why they are needed. In addition, the current descriptions of these sections are difficult to follow. For example, why are the instantiation of explicit compound rules relevant for this task - especially when they are not even used in Equations (1), (2), and (3)? Why is a GAT required to select model-labeled data having inference features consistent with the seed rules? How does such a step ensure diversity, as the Introduction section claims?

**Reproducibility:**

3: Could reproduce the results with some difficulty. The settings of parameters are underspecified or subjectively determined; the training/evaluation data are not widely available.

**Reviewer Confidence:**

4: Quite sure. I tried to check the important points carefully. It's unlikely, though conceivable, that I missed something that should affect my ratings.

---

> ### Author Rebuttal · Authors · 2023-08-29
>
> We sincerely appreciate the reviewer for the constructive and insightful feedback. We are encouraged that the reviewer agrees that our paper is well-motivated, the framework description is ingenious, and the qualitative and quantitative analysis is meaningful. We will explain your concerns point by point.
>
> &nbsp;
>
> #### **R1: One word or several words for reasoning words**
> Thanks for raising this concern. As introduced in Line 306-309 and Figure 3, each type of reasoning words (which is denoted as $w_i$) contains 2-4 single words （depending on the number of mask tokens in the corresponding template). And the embedding of $w_i$ (denoted as $e(w_i)$) is the concatenation of those single words embedding. Then, as illustrated in Line 333-336, the initial node embedding for a data point is $\lbrace w_1||w_2||w_3||w_4\rbrace$, where "$||$" denotes concatenation. We will further clarify this in the revision.
>
> #### **Q1: Time complexity for rules composition evaluation**
> Thanks for your question. Let N be the number of data for rules generation. The time complexity for rules composition evaluation is $O(N)$. Since different data's reasoning rules are generated under different contexts, composing them into compound rules may lead to ambiguity and bring noise to data labeling. Thus, as introduced in Line 259-260, we only evaluate the composition of reasoning rules that are generated from the same data. As each data has four reasoning rules, one data requires $4^2 = 16$ times of composition evaluation and a total $16N$ times for all the data.
>
> #### **R2,Q2: The usage of compound rules instantiation**
> Thanks for your question. As introduced in Equations (2), the matching score "$g(.,.)$" is the sum of the similarity scores between the unlabeled data and rules in $p_S$. Since one data could generate four reasoning rules, $p_S$ is denoted as a subset of them to form a compound rule (As introduced in Line 264-266, 280-282). Without the Compound Rules Compiler, we could not know which composition of the four reasoning rules $\lbrace p_1, p_2, p_3, p_4 \rbrace$ contains enough information to infer the specific relation and avoid confusion with other relations. Thus, the instantiation of compound rules is a necessary step for data labeling. We will clarify this more carefully in the revision.
>
> #### **R2: The motivation of choosing GAT to select model labeled data**
> Graph attention network (GAT) could aggregate neighbor nodes' information and pay less attention to the unimportant nodes. Since the prediction from PLM is not always reliable, and thus there are some noisy seed rules whose reasoning words may contain information irrelevant to relation inference, GAT could alleviate the noise's influence and focus on the nodes with inference features that are close to the real important features pointed by seed rules to select reliable model labeled data for new rules generation. We will further clarify this in the revision.
>
> #### **R2: How to ensure the rules diversity**
> Thanks for raising this concern. First, new rules are generated from model-labeled data. Since RE model could generalize beyond the knowledge provided by the rule-labeled training data, diverse data with representation that is different from original seed rules could be utilized for new rules generation (As introduced in Line 110-113).
>
> Second, we apply GAT to model the relevance between model-labeled data and seed rules. For each relation r, only the data with inference features close to r’s seed rules’ centroid and far from other relations’ could be chosen for further new rules generation, which means the newly added rules not only focus on the consistency with the consensus of previous seed rules, but also focus on being distinguishable from other relations’ seed rules (As introduced in Line 122-124).
>
> #### **Q3, Q4: ARIA performance on domain specific datasets and Re-TACRED**
> Thanks for this useful advice, we will add ARIA’s performance under more domain specific datasets in the further version. To show the ARIA’s effectiveness, we compare AIRA with two best baselines in our paper under Chemprot with 10% training data. ARIA achieves Micro F1 1.58% higher than GradLRE and 3.46% higher than MetaSRE, which shows ARIA could transfer to specific domains easily with well performance. Besides, we also evaluate their performance on Re-TACRED: ARIA gets 2.25% greater than GradLRE and 3.97% greater than MetaSRE.
>
> | Methods     | Chemprot | Re-TACRED |
> |-------------|----------|-----------|
> | MetaSRE     | 60.16    | 66.57     |
> | GradLRE     | 62.04    | 68.29     |
> | ARIA (Ours) | 63.62    | 70.54     |
>
> #### **Q3: Five runs with random sampled training data**
> In a semi-supervised setting, since many works report the mean performance under several runs of training and test using different random seeds [1][2][3], we follow these works’ setting for a fair and reliable comparison (As introduced in Line 423-426). What’s more, as reported in the above works and ours, stratified sampling is utilized for sampling seed data to ensure the label distribution is unchanged in different runs.
>
> #### **Q4: Difference between BERT (Row 3) and BERT w/gold in Table 2**
> Thanks for your question. BERT in the 3rd row of Table 2 refers to the supervised baseline that takes BERT as the encoder and trains on only seed data (as introduced in Line 391-397), while BERT w/gold trains our RE model with both seed data and unlabeled data with gold labels (As introduced in Line 419-421). We will declare their difference more clearly in the future version.
>
> #### **Q4: Greater improvement on TACRED than SemEval**
> Thanks for raising an important point. ARIA could show its advantages more obviously on complex datasets. As introduced in Line 385-389, there are more types of relations with larger label distribution bias in TACRED than in SemEval, making it a dataset that is close to real scenario and require better relation comprehension ability.
>
> First, our framework utilizes PLM's broad knowledge to reason rules that derive the key information related to relation inference and compose them into compound rules for fine-grained labeling among various relations. Especially in the original TACRED dataset there are many incorrectly labeled data, and they are very likely to be labeled with their actual labels by our comprehensive rules.
>
> Second, for each relation, our Graph-based Data Filter could continuously discover data with distinguishable features to generate new rules and enrich the rules set. This could improve the rules' coverage and lead to comprehensive learning for those relation with few seed data.
>
> &nbsp;
>
> [1] Gradient Imitation Reinforcement Learning for Low Resource Relation Extraction
>
> [2] Learning Dual Retrieval Module for Semi-supervised Relation Extraction
>
> [3] Semi-supervised Relation Extraction via Incremental Meta Self-Training

---

### Official Review · Reviewer_WGwB · 2023-08-03

**Soundness:** 2

**Excitement:**

2: Mediocre: This paper makes marginal contributions (vs non-contemporaneous work), so I would rather not see it in the conference.

**Missing References:**

   a) Robust Data Programming with Precision-guided Labeling Functions. AAAI 2020: 3397-3404

   b) Learning to Robustly Aggregate Labeling Functions for Semi-supervised Data Programming. ACL (Findings) 2022: 1188-1202

   c) Semi-Supervised Data Programming with Subset Selection. ACL/IJCNLP (Findings) 2021: 4640-4651

   d) Learning from Rules Generalizing Labeled Exemplars. ICLR 2020

   e) Snuba: Automating Weak Supervision to Label Training Data, VLDB 2018

   f) Automatic Rule Induction for Efficient Semi-Supervised Learning, EMNLP 2022

**Paper Topic And Main Contributions:**

The paper proposes an approach for relation extraction using pre-trained language models (PLM) for creating rules. It uses an iterative approach in which small labeled set and meta-rule templates for creating subsequent rules automatically. These rules are converted into compound rules which are then fed to the rule aggregation algorithm.

**Questions For The Authors:**

The  key motivation seems to enforce reasoning, however, I cannot understand the motivation for their chosen methodology. For instance, the necessity of creating compound rules.

**Reasons To Accept:**

1. The paper proposes an interesting approach for enforcing reasoning using PLMs.

**Reasons To Reject:**

1. The paper is very difficult to follow and understand.
2. Paper misses several relevant weak-supervision and relation extraction baselines . Refer missing citations section.
3. No semi-supervised rule-based baselines are considered.


**Reproducibility:**

1: Could not reproduce the results here no matter how hard they tried.

**Reviewer Confidence:**

5: Positive that my evaluation is correct. I read the paper very carefully and I am very familiar with related work.

---

> ### Author Rebuttal · Authors · 2023-08-29
>
> We sincerely appreciate the reviewer’s helpful feedback. We are encouraged that the reviewer finds our idea interesting and novel. We will explain your concerns point by point.
>
> &nbsp;
>
> #### **R1：Difficult to follow and read**
> Thank you for your suggestion. We will carefully polish our paper in the next version.
>
> #### **R2, R3: Missing baselines for weak-supervision and semi-supervision**
> Thanks for the nice suggestions. Among the baselines mentioned in Missing Reference, the work a) and d) require human-written labeling functions as rules for weak labels, while our rules are generated by PLM from data automatically, which needs less expert knowledge.
> Besides, for the rest of the baselines that could generate rules automatically, we also compare our framework with one of the best baselines, ARI, in our low-resource setting. With 5% training data in SemEval, our framework achieves F1 score **12.37%** higher than ARI, which demonstrates that the comprehensive compound rules distilled in our framework could utilize PLM’s broad knowledge to achieve a more fine-grained data labeling.
>
> | Methods            | Micro F1 (%) |
> |--------------------|--------------|
> | ARI (Ngram+linear) | 67.87        |
> | ARIA (Ours)        | 80.24        |
>
> #### **Q1: Necessity of creating compound rules**
> Thanks for your question. As introduced in Line 105-109, motivated by the work[1], which reported that combining the reasoning process in different ways could lead to a more correct answer, we compose the reasoning rules into compound rules. Besides, in Line 509-517 in Case Study, we also illustrate by examples that a single type of reasoning rule is not enough for relation discrimination. What’s more, in Line 479-488 in the Ablation Study, removing the Compound Rules Compiler may lead to a noise rule labeling and a lower RE model performance. We will further clarify this more clearly in the revision.
>
> [1] Bert: Pre-training of deep bidirectional transformers for language understanding.

---

### Official Review · Reviewer_g2bw · 2023-08-05

**Typos Grammar Style And Presentation Improvements:** Better to add ChatGPT's ICL results t…
**Soundness:** 4

**Excitement:**

3: Ambivalent: It has merits (e.g., it reports state-of-the-art results, the idea is nice), but there are key weaknesses (e.g., it describes incremental work), and it can significantly benefit from another round of revision. However, I won't object to accepting it if my co-reviewers champion it.

**Paper Topic And Main Contributions:**

This paper proposes a rule distillation framework for relation extraction. The framework is composed of (1) a reasoning rule generator based on predefined templates; (2) a compound rule compiler that combines multiple rules; (3) a relation extractor that is trained on both supervised and rule-labeled data. Experiments on two sentence-level RE datasets in the low-resource setting show the effectiveness of the proposed framework.

**Questions For The Authors:**

1. In the soft matching part, why not use NLI to calculate the score? Since the rule is reasoned from data, NLI may be better than embedding similarity.
2. There are many no-relation instances in RE datasets. How do you label them using rules?

**Reasons To Accept:**

1. The paper is well-written and well organized.
2. The proposed framework is interesting incremental work based on existing rule-based weak supervision methods.

**Reasons To Reject:**

1. In experiments, it is necessary to compare with other key methods for low-resource RE, including (1) prompt-based methods (e.g., Knowprompt), and (2) indirect supervision based methods (e.g., NLI for RE, summarization for RE).
2. Some design choices are not well explained (see questions below).

After rebuttal: the authors have added necessary baselines, so I raised the soundness score.

**Reproducibility:**

3: Could reproduce the results with some difficulty. The settings of parameters are underspecified or subjectively determined; the training/evaluation data are not widely available.

**Reviewer Confidence:**

4: Quite sure. I tried to check the important points carefully. It's unlikely, though conceivable, that I missed something that should affect my ratings.

---

> ### Author Rebuttal · Authors · 2023-08-29
>
> We sincerely thank you for the valuable comments. We are encouraged to see that our work is recognized as well-written, well-organized, and novel. We will explain your concerns point by point.
> ####
> #### **R1: Comparison with prompt-based methods and NLI/summarization for RE**
> Thank you for your constructive advice. To show our framework’s effectiveness, we compare ARIA with KnowPrompt[1] and NLI for RE[2] on TACRED with 10% training data. Notice that Roberta is taken as the pre-trained language model for all the above baselines, and stratified sampling is utilized to sample the seed data (Line 423-426). The results are shown as follows:
>
> | Methods       | Micro F1 |
> |---------------|----------|
> | KnowPrompt    | 58.24    |
> | NLI (Roberta) | 59.89    |
> | ARIA (ours)   | 60.86    |
>
> Our work achieves 2.62% F1 scores greater than KnowPrompt and 0.97% greater than NLI for RE. One reason is that these frameworks could not utilize the knowledge from unlabeled data, while our framework continuously discovers high-quality rules from the unlabeled data. What’s more, our work doesn’t require finetuning or prompt-tuning for PLM, which means the unopen-source PLM (like ChatGPT) could be embedded in our framework easily. We will add more comparisons in the future version.
>
> #### **Q1: Whether to use NLI for matching score calculation**
> Thanks for your nice suggestion. With quick computation, cosine similarity is commonly used for soft sentence-rule matching and serves as the maximum inner-product searching in the large-scale retrieval system. Thus, we take cosine similarity for matching score calculation. We also conducted a brief experiment to evaluate the rules’ labeling performance on the same set of 1000 unlabeled data from SemEval with a pre-trained NLI model (Roberta-Large-MNLI) or cosine similarity as the matching manner.
>
> | Methods            | Precision (%) | Recall (%) | Micro F1 (%) |
> |--------------------|---------------|------------|--------------|
> | Cosine Similarity  | 65.17         | 62.29      | 63.69        |
> | Roberta-Large-MNLI | 75.38         | 58.19      | 65.67        |
>
> The NLI model pre-trained on MNLI dataset achieves Micro F1 1.98% higher than cosine similarity in data labeling. This is because the NLI model could focus on not only word-level information, but also the context representation between rules and unlabeled data. We will discuss the NLI model’s potential in our framework in the next version.
>
> #### **Q2：Rules labeling for no-relation instance**
> The data that fails to match with any rules is labeled as "no-relation". In our rules-matching manner, a threshold is set to filter out those data having inconsistent features with rules (As illustrated in Line 292-294). To deal with "no-relation", for unlabeled data, if it is not matched with any compound rules, which means it does not belong to any of the valid relation types, it will be labeled as" no-relation". Besides, for those data that are erroneously labeled as" no-relation" in the original dataset, our rules have the potential to label them with their actual relations.
>
> #### **Typos Grammar Style And Presentation Improvements**
> Thanks for your suggestions. We will move ChatGPT's corresponding results to Table 2 in the next version.
>
> &nbsp;
>
> [1] KnowPrompt Knowledge-aware Prompt-tuning with Synergistic Optimization for Relation Extraction
>
> [2] Label Verbalization and Entailment for Effective Zero and Few-Shot Relation Extraction

---

### Meta-Review · Area_Chair_NWXG · 2023-09-20

**Recommendation:** 3

**Metareview:**

The paper proposes a rule distillation framework for relation extraction and comprises three components: (a) a reasoning rule generator, (b) a compound rule compiler to combine multiple rules, and (c) a relation extractor trained on both supervised and rule-labeled data. The proposed method is evaluated on sentence-level relation extraction in a low-resource setting.

The paper is moderately well-written and represents an incremental advancement in existing rule-based weak supervised methods. However, it lacks some references and semi-supervised baselines. Following discussions, the model was subsequently evaluated on two new datasets, Chemprot and Re-TACRED, as the original dataset TACRED was found to contain incorrect labels. The results indicate a slight performance improvement compared to the current baselines, and this could be further strengthened through statistical significance testing.

---

### Decision · Program_Chairs · 2023-10-07

**Decision:**

Accept-Findings

**Comment:**

The paper proposes a rule distillation framework for relation extraction and comprises three components: (a) a reasoning rule generator, (b) a compound rule compiler to combine multiple rules, and (c) a relation extractor trained on both supervised and rule-labeled data. The proposed method is evaluated on sentence-level relation extraction in a low-resource setting.

The paper is moderately well-written and represents an incremental advancement in existing rule-based weak supervised methods. However, it lacks some references and semi-supervised baselines. Following discussions, the model was subsequently evaluated on two new datasets, Chemprot and Re-TACRED, as the original dataset TACRED was found to contain incorrect labels. The results indicate a slight performance improvement compared to the current baselines, and this could be further strengthened through statistical significance testing.